# Impact of Starch Coating Embedded with Silver Nanoparticles on Strawberry Storage Time

**DOI:** 10.3390/polym14071439

**Published:** 2022-04-01

**Authors:** Ibrahim M. Taha, Ayman Zaghlool, Ali Nasr, Ashraf Nagib, Islam H. El Azab, Gaber A. M. Mersal, Mohamed M. Ibrahim, Alaa Fahmy

**Affiliations:** 1Department of Food Science and Technology, Faculty of Agriculture, Al-Azhar University, Nasr City, Cairo 11884, Egypt; ayman11nasr@gmail.com (A.Z.); aliq11150@azhar.edu.eg (A.N.); ashraf_nagib521@azhar.edu.eg (A.N.); 2Department of Food Science and Nutrition, College of Science, Taif University, P.O. Box 11099, Taif 21944, Saudi Arabia; i.helmy@tu.edu.sa; 3Department of Chemistry, College of Science, Taif University, P.O. Box 11099, Taif 21944, Saudi Arabia; gamersal@tu.edu.sa (G.A.M.M.); ibrahim@tu.edu.sa (M.M.I.); 4Department of Chemistry, Faculty of Science, Al-Azhar University, Nasr City, Cairo 11884, Egypt

**Keywords:** starch, silver nanoparticles, coating, strawberry, shelf-life, washing process

## Abstract

The strawberry has a very short postharvest life due to its fast softening and decomposition. The goal of this research is to see how well a starch-silver nanoparticle (St-AgNPs) coating affects the physical, chemical, and microbiological qualities of strawberries during postharvest life. Additionally, the effect of washing with running water on silver concentration in coated strawberry fruit was studied by an inductively coupled plasma-optical emission spectrometer (ICP-OES). Furthermore, the shelf-life period was calculated in relation to the temperature of storage. Fourier transform infrared-attenuated total reflectance (FTIR-ATR), UV-Visible, and Transmission Electron Microscopic (TEM) were used to investigate the structure of starch-silver materials, the size and shape of AgNPs, respectively. The AgNPs were spherical, with an average size range of 12.7 nm. The coated samples had the lowest weight loss, decay, and microbial counts as compared to the uncoated sample. They had higher total acidity and anthocyanin contents as well. The washing process led to the almost complete removal of silver particles by rates ranging from 98.86 to 99.10%. Finally, the coating maintained strawberry qualities and lengthened their shelf-life from 2 to 6 days at room storage and from 8 to 16 days in cold storage.

## 1. Introduction

Fresh fruit lose quality during postharvest operations such as transportation and storage due to different physiological reactions and processes. The surrounding environment has a direct impact on these changes, which results in the loss of water, texture, color, and nutrients. Therefore, product quality suffers, resulting in lower sale prices and losses for growers [1]. Strawberries (Fragaria ananassa) are a popular fruit all over the world due to their luscious flesh and distinct flavor [2]. This fruit is appealing to eat because it is high in nutrients that are beneficial to human health, specifically vitamins, dietary fibers, anthocyanin, and minerals, among other things [3,4]. Strawberry is a non-climacteric fruit that is particularly perishable. However, physiological problems, fungal infection, and the fragile tissue of the fruit make postharvest handling and storage of fresh strawberries challenging. As a result, from both a technological and economic standpoint, slowing the rate of degradation is a major problem [5].

There are several techniques used as typical postharvest processing procedures for regularly used fruit; among of them is coating. Coating can be defined as a thin layer of a substance that is deposited on food surfaces and acts as a barrier between the food and the environment [6]. Starch is a completely biodegradable polymer, which is abundant as a spare resource in plants with non-toxic and relatively low-cost features. However, it is hard to process due to the high brittleness and limited flexibility [7,8]. It is a semi-crystalline polymer consisting of amylose and amylopectin at different ratios depending on plant sources [9]. Nanomaterials have recently been employed to improve the properties of coating technology by adding nanoparticles into coating matrices [10,11,12]. The efficiency of silver nanoparticles (AgNPs) against a variety of microorganisms has been previously established [13]. AgNPs were prepared using chemical or physical methods [14]. For decades, AgNPs have been categorized as additives and food contact materials for consumer products [15]. They were also approved for use in food coloring (E174) for decorative external coating of chocolates, confectionary, and in liqueurs in the European Union to GMP [16]. Direct usage of silver, or its use in contact with food, in some countries is forbidden; the European Food Safety Authority states that silver should be in the list of suspicious additives [17]. On the contrary, in the USA it is allowed and described as GRAS by the FDA [18]. In this regard, unique and contemporary applications of AgNPs in the food industries have been introduced. For example, AgNPs were recommended as a new alternative to replace or augment the antibacterial characteristics of SO_2_ due to the possible risks of sulfites to human health [19,20]. Additionally, AgNPs are applied as a functional ingredient to prevent caking and clarify liquids [21].

The effects of AgNPs on shelf-life and food quality have been studied. Bakhy et al. [22] discovered that the addition of AgNPs into the coating of red grapes, apples, sweet green peppers, and tomatoes, reduced weight loss and led to extended shelf-life. Additionally, when meat was dipped into a suspension containing AgNPs a positive improvement in the quality of the meat was noted [23]. In accordance with Ortiz-Duarte et al. [24], a coating containing AgNPs reduced the counts of psychrophiles, enterobacteria, yeasts, and molds, while also preserving quality characteristics of fresh-cut melon and extending the shelf-life up to 13 days compared to the uncoated sample. In a more recent study, Saravanakumar et al. [25] noticed that coating with AgNPs and polyvinyl pyrrolidone extended the fresh-cut red and yellow peppers’ shelf-lives by up to 12 days at 4 °C while maintaining all quality parameters of fresh-cut peppers. Likewise, Shahat et al. [26] discussed that the AgNPs-coated apricots could maintain good quality for 24 days in the refrigerator and for 8 days at room temperature.

The goal of this study was to synthesize and characterize AgNPs, as well as investigate the impact of storage duration on strawberries as a food model in the presence of starch–silver nanoparticles, based on some of the justifications described above. Additionally, this study examined the physical, chemical, microbiological, and shelf-life differences between the coated and uncoated strawberry fruit.

## 2. Materials and Methods

### 2.1. Materials

Silver nitrate (assay > 99.9%) was obtained from Merck and maize starch was purchased from the Egyptian Company for Starch and Glucose manufacture, Cairo, Egypt. The remaining chemicals were received from El-Nasr Chemicals Company, Cairo, Egypt.

### 2.2. Methods

#### 2.2.1. Synthesis of Starch-Silver Nanoparticles (St-AgNPs)

In an alkaline medium, St-AgNPs were prepared by using starch as a reducing and capping agent [27]. A total of 1.0 g starch was dissolved in 80 mL of distilled water containing 2.0 g sodium hydroxide using a high-speed homogenizer. Then, the temperature was raised to 60 °C. At this moment, 20 mL of silver nitrate solution was added drop by drop and the reaction was stirred constantly for 60 min. The solution was allowed to cool to room temperature once the reaction was completed. In total, 100 mL of absolute ethanol were added slowly to the above solution with a high-speed homogenizer to precipitate the nanoparticles. The solutions were then centrifuged for 15 min at 4500 rpm. The precipitate was washed (first with ethanol/water (80/20) to remove the unreacted components and byproducts such as NO_3_ ions, then with 100% alcohol). The precipitate was separated, dried, and then characterized as starch-silver nanoparticles (St-AgNPs).

#### 2.2.2. Characterization Methods

##### Visual Examination

As detailed by Anjum et al. [28] the lowering of silver ions was essentially tracked by visual examination of the solutions.

##### Transmission Electron Microscopic (TEM)

The morphological characteristics and size of the produced St-AgNPs were assessed using TEM at a voltage of 200 kV. (2100, JEOL Ltd., Tokyo, Japan). The nanoparticle samples were analyzed by drying a drop of St-AgNPs on a copper grid coated with carbon under ambient conditions.

##### Fourier Transform Infrared (FTIR)

The chemical composition of starch powder was analyzed in comparison to the centrifugated starch/Ag nanocomposite residue with a Fourier transform infrared (FTIR) spectrophotometer Vertex 70 Bruker Co., Ettlingen, Germany in the ATR mode. The spectra were recorded from 4000 to 400 cm^−1^ where the resolution was 4 cm^−1^ with an average of 16 scans.

##### UV-Visible Analysis

The UV-Visible assessment of the synthesized starch/Ag nanocomposites solution was verified employing a UV-Visible spectrophotometer (JASCO V-650, JASCO International, Co., Ltd., Tokyo, Japan) in the wavelength range of 300–700 nm.

##### Antioxidant Activity

The antioxidant activity of AgNPs (at different concentrations of 25, 50, 75, and 100 μg/mL) was estimated using the 1, 1-diphenyl-2-picrylhydrazyl (DPPH) radical scavenging method compared to ascorbic acid as a standard antioxidant at the same concentrations [29].

#### 2.2.3. Starch-AgNPs Application as Active Coatings in the Preservation of Strawberries

Strawberries were bought from a local fruit market in Nasr City (Cairo, Egypt). The fruit were chosen based on the same size, color, absence of physical damage, and fungal breakdown. St-AgNPs (250 and 500 mg·L^−1^) and glycerol as a plasticizer (1.5% *v/v*) were used to prepare coating solutions. Prior to use, the solution was swirled for 15 min at room temperature. The samples were coated using the soaking method according to the method described by Ali et al. [30]. The fruit was rinsed in water and allowed to dry aerobically before being immersed in a starch coating containing AgNPs for 3 min. All samples were packaged in foam trays (125 × 80 × 40 mm), then wrapped with polypropylene stretch film with venting holes of 20 µm thickness. Following that, the samples were separated into two sections, one kept at room temperature (25 ± 3 °C) and the other at cold temperature (6 ± 2 °C). The fruit were tested every two days throughout room storage and every four days during cold storage until spoilage occurred.

#### 2.2.4. Quality Criteria of Strawberry Fruit

##### Loss of Weight

The weight loss (%) was estimated according to Nguyen et al. [31] as follows:
Weight loss (%) = (starting weight − weight at sample date/starting weight) × 100

##### Inspection of Visual Decay (%)

Decay (%) was calculated according to Li et al. [32] as follows:
Decay (%) = (Number of affected fruit at specified storage period/initial number of stored fruit) × 100.

##### Soluble Solids Content (%)

Soluble solids content (TSS) was determined using a hand refractometer and expressed as a % in accordance with AOAC [33].

##### Total Titratable Acidity (%)

Hajji et al. [34] estimated titratable acidity (%) as g of citric acid per 100 g of fresh fruit samples using the formula:
Total titratable acidity (%) = (V (NaOH) × 0.1× 0.064/m aliquot) × 100
where, V: the volume of 0.1 molar NaOH that was used for titration; 0.064 is the conversion factor for citric acid; and m aliquot represents the mass of sample used for titratable acidity determined.

##### Anthocyanin Content

Anthocyanin pigments were extracted using the method of Holzwarth et al. [35] with acidified methanol (0.1% HCl, *v/v*), absorbance was measured at 535 nm using a spectrophotometer, and the total anthocyanin was calculated [36].

##### Induced Coupled Plasma Optical Emission Spectrometry

The operating conditions for measuring silver ions with the Agilent 5100 Synchronous Vertical Dual View inductively coupled plasma-optical emission spectrometer (ICP-OES, Vista Pro, Varian, Mulgrave, Australia)are presented in Table 1.

The Anton-Paar microwave digestion device (multi-wave PRO) was employed to digest strawberry fruit in an acid solution. The solutions were digested to create an adequate matrix for quantifying silver ions and to ensure a sufficient and consistent recovery that was compatible with the analytical techniques of APHA et al. [37].

##### Microbial Counts

According to the techniques given by APHA [38], the microbial loads for testing strawberry samples were counted using nutritional agar for bacteria and potato dextrose agar for yeasts and molds. Bacteria were incubated for 24–48 h at 37 °C, while yeasts and molds were incubated for 3–5 days at 25 °C. Log CFU/g was used to express all microbial counts.

##### Strawberry Shelf-Life

The strawberry shelf-life was determined by calculating the number of days required for them to remain marketable [39].

##### Analytical Statistics

The statistical analysis was carried out using the Co-statistical software package [40] by analyzing the variance (one way completely randomized, ANOVA). The differences between treatments means were calculated using Duncan’s multiple range test with *p* < 0.05 as the significance level.

## 3. Results

### 3.1. Characterization Results

#### 3.1.1. Nanoparticle Visual Examination

The reaction media created a clear yellow color with the addition of AgNO_3_ solution to starch solution with stirring, then turned to brown, then dark brown, with mirror-like lighting on the sides of the Erlenmeyer flask plainly indicating the creation of St-AgNPs in the reaction media.

#### 3.1.2. Morphology and Microstructure

The morphology of the prepared nanocomposite (starch-AgNPs) was studied using transmission electron microscopy (TEM) and is shown in Figure 1. It can be seen that the St-AgNPs were successfully fabricated as spherical nanoparticles with an average diameter of 12.7 nm (inset of Figure 1). It is worth mentioning that the starch molecules act as a capping and reducing agent.

#### 3.1.3. Chemical Composition of Nanocomposite

Starch consists mainly of amylose and amylopectin which include mostly OH and C–O–C groups in addition to the CH and CH_2_ backbone. The FTIR spectra of pure starch compared to starch-AgNPs are presented in Figure 2.

The FTIR spectrum of starch reveals characteristic absorption bands at ~1080 and ~850 cm^−1^ (C–O–C symmetrical stretching), ~1000 and ~928 cm^−1^ (COH bending), ~1148 cm^−1^ (C–O stretching), ~1246 cm^−1^ (deformation vibration of C–O–H), ~1644 cm^−1^ (OH bending of water), and ~3310 cm^−1^ (OH stretching) [41,42]. However, the FTIR spectrum of starch-AgNPs displays a similar form to that of pure starch [43], but there is an obvious difference in the intensity of bands at ~3310, 1450, and 1000 cm^−1^ before and after the reaction of silver nitrate with starch. The intensity of bands at ~3310 and 1000 cm^−1^ was reduced in starch-AgNPs. This reducing can be signified as the intramolecular hydrogen bonding produced from the formation of a novel interaction between the OH-groups in starch and AgNO_3_ as OH⋅⋅⋅AgNO_3_. Furthermore, starch has many active OH-groups and these groups are capable of interacting with incorporated Ag^+^ ions via chemical bonds and steric entrapment. These connections of the starch layer with Ag^+^ leads to the involvement of Ag^+^ with the polymer layer. Therefore, the possible formation of silver nanoparticles as an in-situ reduction of silver ions to the metallic state should be considered [44]. On the other hand, the beak that was observed at ~1450 cm^−1^ extensively increased in intensity and might be attributed to the proton transfer from the –OH groups to the NO_3_ groups, indicating the formation of ionic interactions.

#### 3.1.4. Investigation of St-AgNPs Formation Using UV-Visible Spectroscopy

Ultraviolet-visible absorption spectroscopy (UV-Vis) was employed to support the AgNPs formation and study the effect of AgNPs embedded in the starch matrix. The most specific indication of AgNPs formation is the surface plasmon resonance (SPR) bands recognizable in the region of 350–600 nm [45,46]. The colloidal dispersions were reserved in a dark place, avoiding the photochemical reactions. UV-Vis findings were achieved on the colloidal dispersion AgNPs (0.98 g·L^−1^) in the starch matrix solution (1%) in comparison to pure starch solution.

As shown in Figure 3 an absorption peak at 410 nm was observed, indicating the SPR of AgNPs and agreeing well with the data reported by [47,48]. Moreover, the solution does not contain many aggregated particles; therefore, the plasmon band was found in a symmetric manner.

#### 3.1.5. Antioxidant Activity of AgNPs

The 1, 1-diphenyl-2-picrylhydrazyl (DPPH) assay was used to evaluate the antioxidant activity of produced St-AgNPs compared to that of ascorbic acid, as shown in Figure 4. The antioxidant activity of St-AgNPs was lower than ascorbic acid at the same concentrations (see Figure 4).

The scavenging activity (%) increased from 25 to 59% for St-AgNPs and from 48 to 71% for ascorbic acid with increasing concentrations from 25 to 100 mg·L^−1^. Generally, the above-mentioned data exhibited good antioxidant potency and suggested the possibility that St-AgNPs-based coatings could be effectively employed as new antioxidant materials for application in food processing.

### 3.2. Quality Criteria of Strawberry Fruit

#### 3.2.1. Weight Loss

Strawberry fruits are prone to rapid water loss, resulting in an extremely thin skin and tissue shrinkage, as well as fragility. Table 2 shows the effect of starch coatings containing AgNPs on weight loss (%) during storage. A progressive increase in weight loss % was noted in all strawberry samples at the various temperatures throughout extended storage periods, which could be connected to the ongoing transfer of water from the fruit’s surface to the external environment [49]. The weight loss of the uncoated sample was greater than coated samples in all stages of storage at room temperature. After two days of storage, the uncoated sample lost 5.8% of initial weight compared to 4.5 and 4.3% for coated samples with St-AgNPs at 250 and 500 mg·L^−1^, respectively. After six days of storage, the uncoated sample showed a 28% loss in weight; meanwhile, the strawberries coated with St-AgNPs 250 and 500 mg·L^−1^ exhibited 16.9 and 15.8%, respectively, with significant differences.

In terms of the effect of storage at 6 ± 2 °C, coating techniques clearly minimized weight loss in treated strawberries in comparison to the untreated fruit. There was no significant difference in weight loss between coated strawberry samples up to day 12 of storage. Differences were discovered among coated samples enduring the final days of storage, with the highest losses recorded with uncoated strawberries (13.6%); while, samples coated with the St-AgNPs at 250 and 500 mg·L^−1^ recorded losses of 5.7 and 4.4%, respectively.

The findings were generally consistent with prior research that showed a reduction in weight loss that could be attributed to the coatings, where these coatings act as partially permeable barriers to gas and water, lowering respiration, moisture loss, and oxidation reactions. Additionally, the nanoparticles are responsible for forming crisscrosses inside the coating layer; thus preventing penetrates such as oxygen, carbon dioxide, and vapor from passing through it [50].

#### 3.2.2. Visual Decay

One of the primary causes of postharvest losses is decay. Table 3 shows the effect of a starch coating containing AgNPs on the percentage of strawberry fruit that deteriorates. Throughout storage time, the quantity of deterioration increased dramatically in all samples. Coating treatments, on the other hand, lowered the proportion of deterioration. These findings are consistent with those of Khodaei et al. [51] who showed that coating the strawberries reduced visible deterioration when compared with the uncoated sample.

During storage at room temperature, visual decay on uncoated strawberries began quickly on the second day with 10% and the rate of infection was increased with the increased period of storage reaching 80% after two additional days of storage (day 4), and 100% on the sixth day. On the other hand, the infection percentage for fruit coated with St-AgNPs 250 mg·L^−1^ was 20% on day 4 of storage and it was 60% on day 6. Whereas the fruit coated with St-AgNPs 500 mg·L^−1^ (did not show signs of infection on the fourth day of storage) showed a 40% infection by the end of storage (day 6).

During cold storage, uncoated sample fruit showed evidence of degradation (15%) on the fourth day, and 45% became degraded after an additional four days (day 8). The coated strawberries exhibited no signs of apparent decay until the eighth day of storage. On day 12, decay rates in uncoated fruit climbed to 100%, while the decay in coated fruit showed rates of 20 and 10%, respectively, for St-AgNPs 250 and 500 mg·L^−1^. At the end of storage (day 16) 50% of strawberries coated with 250 mg·L^−1^ St-AgNPs and 30% of fruit coated with 500 mg·L^−1^ St-AgNPs were infected. The antibacterial activity of St-AgNPs may have contributed to their superior coating efficacy against spoiling.

#### 3.2.3. Total Soluble Solid Content

Soluble solids content (TSS) has a large influence on fruit qualities and consumer perceptions. Table 4 shows the TSS content of strawberry samples as a function of St-AgNPs coatings during storage at room or cold temperatures. Strawberry samples maintained at 25 ± 3°C had greater TSS% than those preserved at 6 ± 2 °C. No significant differences in the effect of coating between samples during storage except for the second day at room temperature were observed. In addition, the TSS of all samples increased during storage periods at all temperatures. These increase throughout storage, owing to the degradation of starch to soluble sugars or cell membrane hydrolysis.

During the storage at 25 ± 3 °C, on the second day, the uncoated sample exhibited a higher increase in TSS compared to those coated, where the incremental rates were 15.4, 6, and 5% for the uncoated strawberries and coated samples with St-AgNPs at 250 and 500 mg·L^−1^, respectively. On day 4, the uncoated sample rejected and the coated one with St-AgNPs (250 and 500 mg·L^−1^) showed an increase to reach 6.2 and 6.1%, respectively. At the end of storage (day 6), TSS contents reached 6.2 and 6.2% for strawberries coated with St-AgNPs at 250 and 500 mg·L^−1^. TSS content for the uncoated sample started to decline on the eighth day, but the TSS for the coated sample with St-AgNPs 250 mg·L^−1^ began to decrease on the 16th day, possibly due to hydrolysis. These findings are consistent with those of Nguyen et al. [52] who discovered that TSS% for strawberries increased before gradually declining due to the hydrolysis at the end of the experiment. For fruit treated with St-AgNPs 500 mg·L^−1^, the TSS increased during storage periods and reached 6.1% at the end of storage.

#### 3.2.4. Total Acidity Content

Table 5 shows the effect of a starch coating containing AgNPs on the total acidity (TA) content of strawberry fruit. In general, acidity contents in coated samples did not differ throughout storage time. Uncoated and coated fruit, on the other hand, displayed differences between them. A progressive drop in acidity was also observed for strawberry samples maintained at various temperatures over time. The decline in TA might be attributed to a shift in fruit metabolism due to the consumption of acids during the respiration process [53].

Regarding storage at room temperature, the acidity content of untreated and treated strawberry samples with the nano-coating (250 and 500 mg·L^−1^ of St-AgNPs) decreased from ~0.5% on the initial day to 0.4% on the second day with a significant difference discovered for the untreated strawberries. At the end of storage (day 6), the acidity content of samples treated with St-AgNPs at 250 and 500 mg·L^−1^ was ~0.3%, meaning that by limiting respiration, the nano-coating can slow down the use of acids in the physiological metabolic reactions of fruit.

The information regarding the storage of strawberry samples at a cold temperature is shown in Table 5. A difference was detected starting on day 4 for the uncoated strawberries. On day 8 of storage, the TA content for uncoated fruit reached 0.35%, while the TA for samples coated with St-AgNPs at 250 and 500 mg·L^−1^ reached 0.42 and 0.43%, respectively. These results are an indication of the efficiency of the coating in maintaining the TA of the tested samples during storage. On day 12, coated strawberries recorded 0.40 with 250 mg·L^−1^ of St-AgNPs and 0.41% with 500 mg·L^−1^ of St-AgNPs, while the uncoated sample was rejected. At the end of cold storage (days 16), the sample coated with St-AgNPs 500 mg·L^−1^ had the smallest decline in TA (8.0%). According to the data, the greater acidity drop in uncoated strawberries could be attributed to the use of acids as precursors for metabolism throughout preservation. Nano-coating can lower fruit respiration and slow acid consumption in the physiological metabolic activities of fruit. As a result, the nano-membrane improves the shelf-life of strawberries.

#### 3.2.5. Content of Anthocyanin

The color of the fruit is the most essential feature for determining their quality. Table 6 summarizes the content of anthocyanin in strawberry fruit throughout storage.

There was no variation in anthocyanin content between the untreated and treated fruit during initial storage, although it did differ significantly throughout the remaining days of storage. Overall, the anthocyanin content of all strawberry fruit increased as storage periods progressed at all temperatures, although it dropped as storage periods for cold storage ended. This could be attributed to increased enzyme (polyphenol oxidase) activity. These findings are consistent with prior research by Riaz et al. [54]. Throughout storage at room temperature, the uncoated strawberries had a greater rise in anthocyanin content (527.6 mg·kg^−1^) than the coated fruit, which had 475.0 and 471.6 mg·kg^−1^ on the second day of storage. Apart from the effect of substantial weight loss, which could add to the concentration of anthocyanin pigments, it could be described as a natural process throughout fruit ripening [55]. Additionally, on the fourth day, the anthocyanin content of samples coated with 250 and 500 mg·L^−1^ St-AgNPs were 572.5 and 562.0 mg·kg^−1^, respectively. These values reached 629.0 and 606.0 mg·kg^−1^ at the end of storage. On the other hand, after four days of storage at cold temperatures, the content of anthocyanin for uncoated and coated samples with St-AgNPs at 250 and 500 mg·L^−1^ increased to 446.8, 371.5, and 363.5 mg·kg^−1^, respectively. On day 8 of cold storage, the uncoated sample had a drop in anthocyanin content (383.6 mg·kg^−1^), but the coated strawberries had a continual increase to 414.5 and 409.5 mg·kg^−1^, respectively. However, at the end of storage (day 16), a gradual decline in anthocyanin was observed and reached 467.0 and 465.6 mg·kg^−1^ for samples treated with St-AgNPs 250 and 500 mg·L^−1^, respectively. Similar findings were observed by Khodaei et al. [51] who found that the concentration of anthocyanin in coated strawberries dropped after storage, with a significant effect between coated and uncoated fruit samples.

#### 3.2.6. Impact of the Washing Process on Silver Concentration

After acid digestion, the silver concentration in the produced silver nanoparticles solution and strawberry fruit samples was determined using ICP-OES. Initially, it was found that the concentration of Ag in the prepared St-AgNPs solution was 9880 mg·Ag·L^−1^ solution.

Figure 5 depicts the influence of the washing process on silver concentrations. As a general trend, the washing process of coated strawberry fruit after storing them at previously mentioned periods led to almost complete removal of silver particles from the fruit samples by rates ranging from 98.86 to 99.10%. The initial readings for strawberries coated with 250 and 500 mg·L^−1^ AgNPs were 1.40 and 2.11 mg·Ag·kg^−1^, respectively. After the washing process, the strawberries stored at room temperature for 6 d recorded 0.016 and 0.020 mg·Ag·kg^−1^ (with reduction ratios 98.86 and 99.05%) for samples coated with 250 and 500 mg·L^−1^, respectively. While the same treatments recorded 0.014 and 0.019 mg·Ag·kg^−1^ (with reduction ratios 99.00 and 99.10%), respectively, after 16 d of cold storage.

Generally, the levels detected in fruit samples after washing treatment were below the oral reference dose for silver in humans of 0.005 mg/kg/d equivalent 0.35 mg·Ag per person (70 kg)/day [56].

#### 3.2.7. Changes in Microbial Load for Strawberry Fruit during Storage

Controlling the microbial load is critical for preserving high qualities, particularly the final product’s sanitary condition [57]. The counts of total bacterial, mold, and yeast were decreased, as indicated in Figure 6 and Figure 7 when the coating was applied. It is worth noting that the initial TBC of all samples increased with increasing storage time, especially in the control fruit that were increased faster than other samples.

At room temperature, the initial levels of TBC increased to 4.37 log CFU/g for the untreated sample, compared to 3.48 and 3.30 log CFU/g for samples treated by St-AgNPs 250 and 500 mg·L^−1^, respectively, on the 2nd day. On day 6 of storage, fruit coated with St-AgNPs 500 mg·L^−1^ had the lowest bacterial count (5.14 log CFU/g). On the other hand, regarding storage in cold conditions (6 ± 2 °C), on day 4 the uncoated strawberry had a rapid increase to 4.96 log CFU/g. Coated samples recorded a modest increase of 3.05 and 2.80 log CFU/g, respectively, for treatments with St-AgNPs 250 and 500 mg·L^−1^, showing the effectiveness of coatings on bacterial growth inhibition. On day 16, fruit coated with St-AgNPs had lower aerobic bacteria count. From the data, the coating was more active in controlling TBC as compared to the control in strawberry fruit. However, silver nanoparticles can strongly suppress the growth of fruit bacteria [58].

Additionally, Figure 7 depicts the changes in yeast and mold values in treated and untreated strawberries over time. The initial levels of yeast and mold found in fresh fruit were 2.28 log CFU/g; however, immediately after the coating treatment, the levels of yeast and molds were decreased to 2.26 and 2.18 log CFU/g, respectively. These findings agree well with those found by Shahbazi et al. [59], indicating 2.11 log CFU/g for the yeast and mold count for fresh strawberries. As shown in Figure 7 the yeast and mold count increased in all samples during storage, with the uncoated sample increasing faster and reaching higher values than the coated samples.

In terms of the impact of storage at room temperature, the untreated strawberry had the greatest yeast and mold count (5.85 log CFU/g) on day 4, while the coated strawberries with St-AgNPs 500 mg·L^−1^ had the lowest yeast and mold count (3.87 log CFU/g). At the end of room storage, mold and yeast counts were 5.35 and 4.98 log CFU/g for samples treated with St-AgNPs at 250 and 500 mg·L^−1^, respectively. On the other hand, regarding storage in the fridge (6 ± 2 °C), on day 16 fruit coated with St-AgNPs at 500 mg·L^−1^ had lower mold and yeast counts (5.44 log CFU/g). Referring to data obtained and presented in Figure 7**,** it can be concluded that nanocoating had antimicrobial effects on the mold and yeast by limiting or delaying their growth. A previous study by Shahbazi [60] found that the coating solution significantly reduced the microbial growth of chilled stored strawberries in comparison with the uncoated samples.

#### 3.2.8. Shelf-Life Periods for Strawberry Fruit

The ability to lengthen the shelf-life of the foods is the most essential benefit of any antibacterial treatment [61]. Strawberries maintained at a cold temperature had a greater shelf-life than those maintained at room temperature overall. The data in Figure 8 shows that coating strawberry fruit can change their rotting profile and lengthen their shelf-life.

The shelf-life of strawberries was determined physically by observing color changes and diseased patches on the surface. For example, after day 2 of storage at room temperature and day 8 at cold temperature, the uncoated fruit deteriorated. The sample coated with 250 mg·L^−1^ St-AgNPs, on the other hand, had a shelf-life of 4 and 12 days when stored at 25 ± 3 and 6 ± 2 °C, respectively. While fruit treated with St-AgNPs 500 mg·L^−1^ had a shelf-life of 6 and 16 days, respectively, when stored at 25 ± 3 and 6 ± 2 °C. For strawberry growers, this is a substantial economic and encouraging result.

## 4. Conclusions

The spherical shape of St-AgNPs was synthesized employing starch as a reducing and capping agent. This approach is simple, quick, cost-effective, has no toxic chemicals, and is environmentally safe. The preparation of St-AgNPs was monitored, with the first sign being a change in the color of the solution. Second, the size distribution of Ag nanoparticles was discovered to be in the 12.7 nm range utilizing TEM images. The application of St-AgNPs to coat strawberries was shown to be beneficial by retaining the quality of the fruit during storage, where the maintained weight, TSS, anthocyanin, and total acidity during storage at 6 ± 2 and 25 ± 3 °C compared to uncoated fruit was investigated. Analyses of silver concentration demonstrated that concentration reduced to 0.014 mg·Ag·kg^−1^ in strawberries using simple washing with running water. Strawberry fruit coated with St-AgNPs could be maintained in good condition for 6 days at 25 ± 3 °C and for 16 days at 6 ± 2 °C, according to the microbiological evaluation. Finally, to make use of nanotechnology’s unique qualities, we advocate encouraging its use in food processing, particularly in food packaging trends.

## Figures and Tables

**Figure 1 polymers-14-01439-f001:**
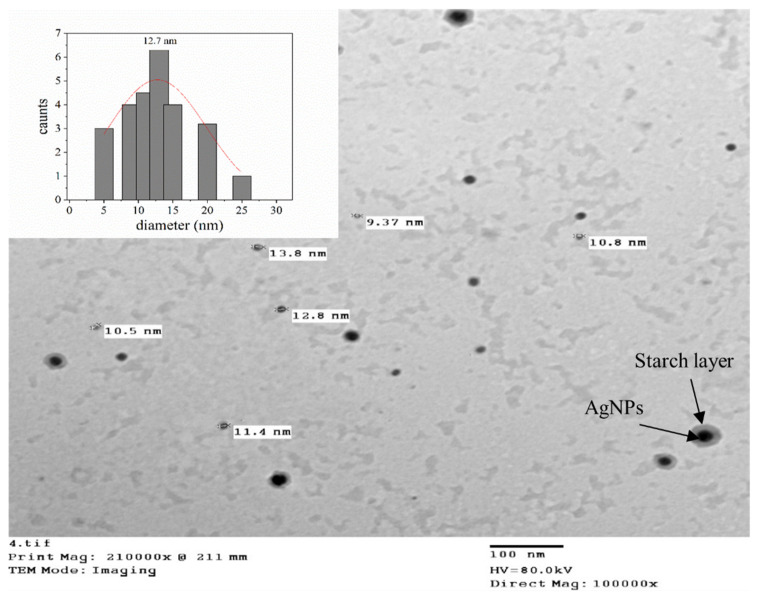
Particle size, shape, and distribution of starch-silver nanoparticles examined using TEM.

**Figure 2 polymers-14-01439-f002:**
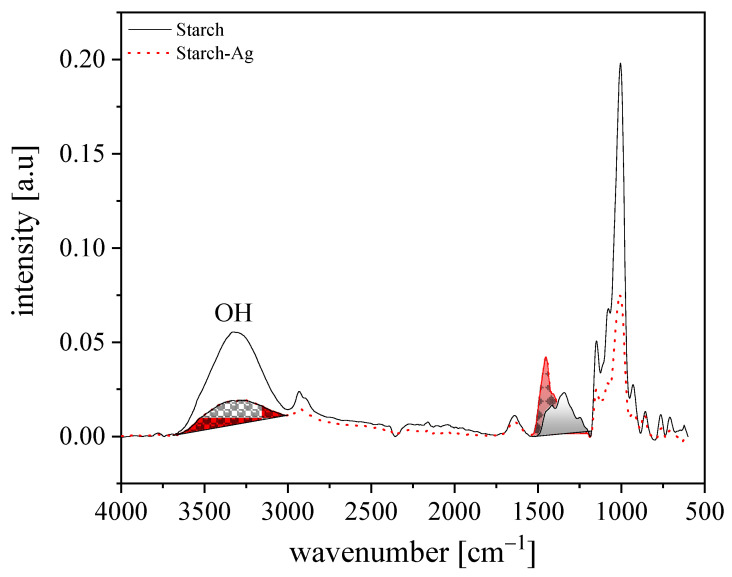
Spectra of starch-Ag nanocomposite in comparison to pure starch as the blank.

**Figure 3 polymers-14-01439-f003:**
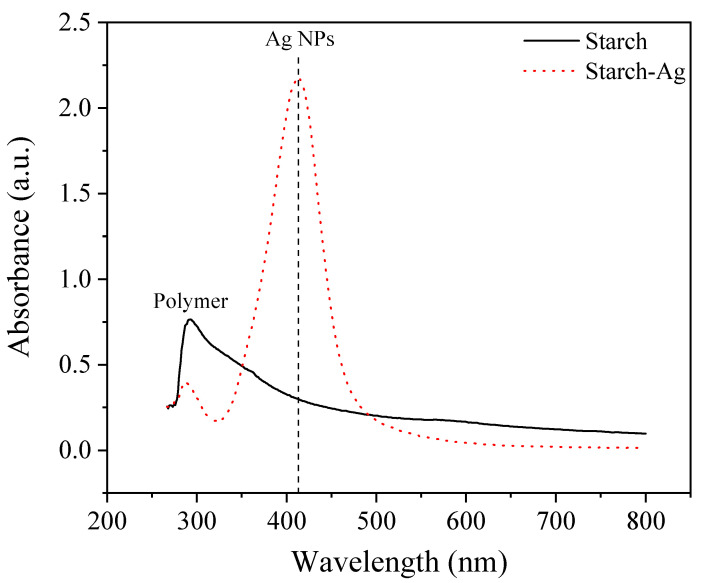
UV-Vis. spectrum of a starch-Ag nanocomposite compared to pure starch.

**Figure 4 polymers-14-01439-f004:**
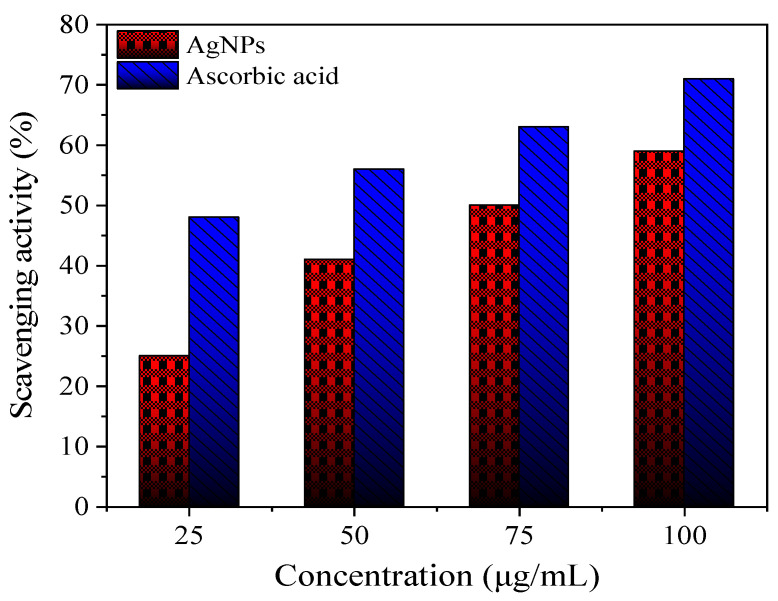
DPPH free radical scavenging activity (%) of St-AgNPs compared to ascorbic acid.

**Figure 5 polymers-14-01439-f005:**
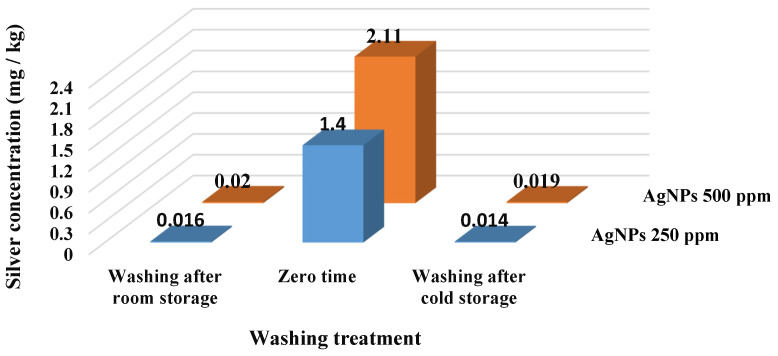
Analysis of the effect of the washing process on silver removal by the inductively coupled plasma-optical emission spectrometer (ICP-OES).

**Figure 6 polymers-14-01439-f006:**
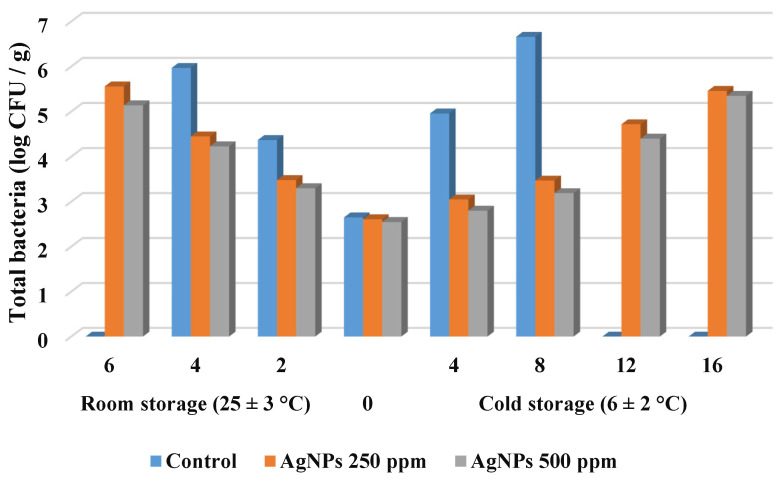
Changes in total bacterial count (log CFU/g) of strawberry fruit during storage.

**Figure 7 polymers-14-01439-f007:**
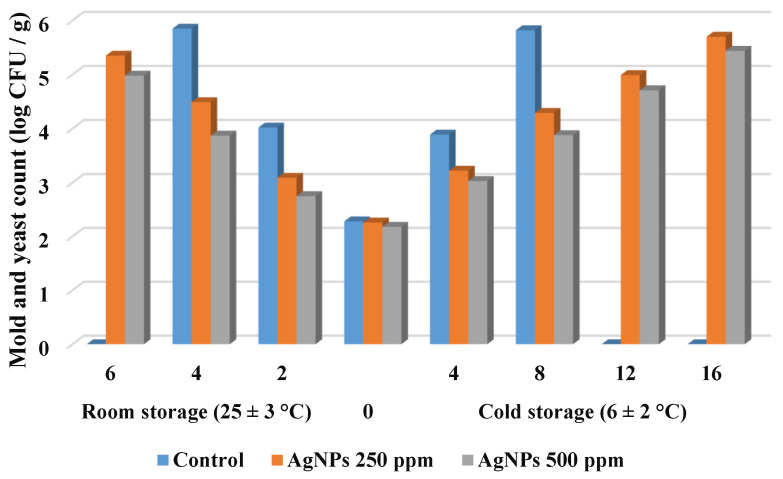
Changes in yeast and mold counts (log CFU/g) of strawberry fruit during storage.

**Figure 8 polymers-14-01439-f008:**
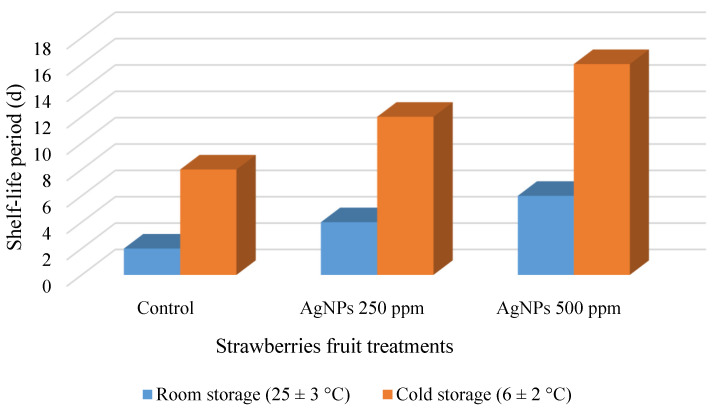
Shelf-life of strawberry fruit.

**Table 1 polymers-14-01439-t001:** Operating conditions for the determination of silver ions by inductively coupled plasma–optical emission spectrometer (ICP-OES).

Operating Conditions	Silver Ions
Wavelength (nm)	328.1
Intensity	82675
RF plasma torches power (kW)	1.2
Nebulizer flow (L/min)	0.7
Plasma flow (L/min)	12
Viewing mode	Axial
Background correction	Fitted
Correlation coefficient	0.98
Calibration error (%)	10
Calibration fit	Rational
Limit of quantification (ug/dl)	5

**Table 2 polymers-14-01439-t002:** Changes in weight loss (%) in strawberry fruit during storage.

Treatment		Storage Period (d) at 25 ± 3 °C			Storage Period (d) at 6 ± 2 °C			
		2	4	6	4	8	12	16
Control		5.80 ^Ca^	11.50 ^Ba^	28.00 ^Aa^	2.35 ^Da^	4.35 ^Ca^	8.35 ^Ba^	13.65 ^Aa^
AgNPs	250 mg·L^−1^	4.15 ^Cb^	7.45 ^Bb^	16.89 ^Ab^	2.21 ^Da^	2.77 ^Cb^	3.40 ^Bb^	5.65 ^Ab^
	500 mg·L^−1^	4. 25 ^Cb^	7.35 ^Bb^	15.75 ^Ac^	2.08 ^Da^	2.65 ^Cb^	3.28 ^Bb^	4.44 ^Ac^

Values in the same row or the same column with different letters are significantly different according to Duncan’s Multiple Range Test at *p* < 0.05. The capital letter refers to the time factor, while the small letter refers to the treatment factor.

**Table 3 polymers-14-01439-t003:** The changes in the percentage of visual decay during storage.

Treatments		Storage Period (d) at 25 ± 3 °C			Storage Period (d) at 6 ± 2 °C			
		2	4	6	4	8	12	16
Control		10.0	80.0	100.0	15.0	45.0	100.0	ND
AgNPs	250 mg·L^−1^	ND	20.0	60.0	ND	ND	20.0	50.0
	500 mg·L^−1^	ND	ND	40.0	ND	ND	10.0	30.0

ND: not determined.

**Table 4 polymers-14-01439-t004:** TSS content of strawberry samples as a function of St-AgNPs coatings during storage.

Treatments		Storage Period (d) at 25 ± 3 °C				Storage Period (d) at 6 ± 2 °C				
		0	2	4	6	0	4	8	12	16
Control		5.70 ^Ba^	6.58 ^Aa^	ND	ND	5.70 ^Aa^	5.87 ^Aa^	5.65 ^Aa^	ND	ND
AgNPs	250 mg·L^−1^	5.70 ^Ba^	6.04 ^Ab^	6.20 ^Aa^	6.18 ^Aa^	5.70 ^Ba^	5.84 ^Ba^	5.95 ^ABa^	6.22 ^Aa^	6.21 ^Aa^
	500 mg·L^−1^	5.65 ^Ba^	5.93A ^Bb^	6.10 ^Aa^	6.20 ^Aa^	5.65 ^Ca^	5.72 ^BCa^	5.80 ^ABCa^	6.02 ^Aa^	6.08 ^Aa^

Values in the same row or the same column with different letters are significantly different according to Duncan’s Multiple Range Test at *p* < 0.05. The capital letter refers to the time factor, while the small letter refers to the treatment factor. ND: not determined.

**Table 5 polymers-14-01439-t005:** Changes in total acidity of strawberries during storage.

Treatments		Storage Period (d) at 25 ± 3 °C				Storage Period (d) at 6 ± 2 °C				
		0	2	4	6	0	4	8	12	16
Control		0.48 ^Aa^	0.37 ^Ba^	ND	ND	0.48 ^Aa^	0.41 ^ABa^	0.35 ^Ba^	ND	ND
AgNPs	250 mg·L^−1^	0. 47 ^Aa^	0.42 ^Aa^	0.38 ^Aa^	0.32 ^Aa^	0.47 ^Aa^	0.45 ^Aa^	0.42 ^Aa^	0.40 ^Aa^	0.37 ^Aa^
	500 mg·L^−1^	0.46 ^Aa^	0.42 ^Aa^	0.39 ^Aa^	0.34 ^Aa^	0.46 ^Aa^	0.47 ^Aa^	0.43 ^Aa^	0.41 ^Aa^	0.38 ^Aa^

Values in the same row or the same column with different letters are significantly different according to Duncan’s Multiple Range Test at *p* < 0.05. The capital letter refers to the time factor, while the small letter refers to the treatment factor. ND: not determined.

**Table 6 polymers-14-01439-t006:** Anthocyanin content (mg·kg^−1^) in strawberry fruit during storage.

Treatments		Storage Period (d) at 25 ± 3 °C				Storage Period (d) at 6 ± 2 °C				
		0	2	4	6	0	4	8	12	16
Control		325.2 ^Ba^	527.6 ^Aa^	ND	ND	325.2 ^Ca^	446.8 ^Aa^	383.6 ^Bb^	ND	ND
AgNPs	250 mg·L^−1^	322.5 ^Da^	475.0 ^Cb^	572.5 ^Ba^	629.0 ^Aa^	322.5 ^Da^	371.5 ^Cb^	414.5 ^Ba^	485.0 ^Aa^	467.0 ^Aa^
	500 mg·L^−1^	324.4 ^Da^	471.6 ^Cb^	562.0 ^Ba^	606.0 ^Ab^	324.4 ^Da^	363.5 ^Cb^	409.5 ^Ba^	471.5 ^Aa^	465.6 ^Aa^

Values in the same row or the same column with different letters are significantly different according to Duncan’s Multiple Range Test at *p* < 0.05. The capital letter refers to the time factor, while the small letter refers to the treatment factor. ND: not determined.

## Data Availability

Not applicable.

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
