# Peer review of "Impact of Starch Coating Embedded with Silver Nanoparticles on Strawberry Storage Time"

_polymers, 2022, doi:10.3390/polym14071439_

Round 1
Reviewer 1 Report
- The abstract has not a good start with this sentence “Due to its fast softening and decomposition, the strawberry has a very short postharvest life”. Better to remove this sentence or use it in the other lines.
- Section 2.2.3 needs a reference.
- The composition of AgNO3 is safe to use? The NO3 composition has some criteria to use. I suggest running an MTT assay on the final product and include in the manuscript.
- Compare all the results of this study with other similar studies. The description is very weak. Please explain the results more.
- The following references are suggested to be used in this manuscript:
Sabbagh, F., Kiarostami, K., Mahmoudi Khatir, N., Rezania, S., & Muhamad, I. I. (2020). Green synthesis of Mg0. 99 Zn0. 01O nanoparticles for the fabrication of κ-Carrageenan/NaCMC hydrogel in order to deliver catechin. Polymers, 12(4), 861.
Sabbagh, F., Kiarostami, K., Khatir, N. M., Rezania, S., Muhamad, I. I., & Hosseini, F. (2021). Effect of zinc content on structural, functional, morphological, and thermal properties of kappa-carrageenan/NaCMC nanocomposites. Polymer Testing, 93, 106922.
- The writing is weak and there are some typo errors. There is a need to be checked by a native.
Author Response
Comments and Suggestions for Authors
- The abstract has not a good start with this sentence “Due to its fast softening and decomposition, the strawberry has a very short postharvest life”. Better to remove this sentence or use it in the other lines.
We appreciate the insightful suggestions concerning this issue; therefore, it was modified to be more convenient.
- Section 2.2.3 needs a reference.
Th reference was added.
- The composition of AgNO3 is safe to use? The NO3 composition has some criteria to use. I suggest running an MTT assay on the final product and include in the manuscript.
We absolutely agree with the reviewer and therefore, the following sentences were highlighted in the manuscript: 100 mL of absolute ethanol was added slowly to the solution with a high-speed homogenizer to precipitate the nanoparticles. The solutions were then centrifuged for 15 min at 4,500 rpm. The precipitate was washed (first with ethanol/water (80/20) to remove the unreacted components and byproducts such as NO3 ions, then with 100 % alcohol).
- Compare all the results of this study with other similar studies. The description is very weak. Please explain the results more.
We are grateful for the reviewer’s input. The results were explained as much we can.
- The following references are suggested to be used in this manuscript:
Sabbagh, F., Kiarostami, K., Mahmoudi Khatir, N., Rezania, S., & Muhamad, I. I. (2020). Green synthesis of Mg0. 99 Zn0. 01O nanoparticles for the fabrication of κ-Carrageenan/NaCMC hydrogel in order to deliver catechin. Polymers, 12(4), 861.
Sabbagh, F., Kiarostami, K., Khatir, N. M., Rezania, S., Muhamad, I. I., & Hosseini, F. (2021). Effect of zinc content on structural, functional, morphological, and thermal properties of kappa-carrageenan/NaCMC nanocomposites. Polymer Testing, 93, 106922.
Both were added in the introduction section.
- The writing is weak and there are some typo errors. There is a need to be checked by a native.
We would like to thank the reviewer for raising this issue. The language in the whole manuscript was revised by a native English speaker and American citizen Mr. Hassan Omar (researcher at 6.6 Physics and chemical analysis of polymers, BAM Berlin, Germany). Therefore, we have mentioned him in the acknowledgment.

Reviewer 2 Report
Dear Author/Editor,
Paper Impact of starch coating embedded with silver nanoparticles on strawberry storage time (1654076) deals with synthesis and characterization of starch-silver nanoparticles and application of this coating on strawberries. Finally, strawberries characterization over time was given regarding weather coating was applied or not and weather strawberries were stored at room temperature and at cold storage.
Paper’s topic is up-to-date, which is sustained with contemporary literature. Title and abstract are appropriate, so as proposed methodology. Experimental set up is clear, results well explained.
I have no significant remarks, except for:
1. The order of methods for starch-silver nanoparticles characterization and quality criteria of strawberry fruit should be in the same order as displayed results. Please uniform the order of methods and results to be the same.
2. Table 1 should have some kind of announcement in the text. For example: Operating conditions for… are presented in Table 1.
3. In part 2.2.3. (lines 139-148) provide additional info about how long immersion lasted; did you repeated the procedure or it was done once? Weather strawberry samples were kept in bulk or packed in some additional packaging material during storage period?
4. Change subtitle 3.2.3. to „Total soluble solid content”
Once again, I acknowledge the quality of the work and propose it for publication after minor revision.
Author Response
Our correspondences to Reviewer #2:
Comments and Suggestions for Authors
Paper Impact of starch coating embedded with silver nanoparticles on strawberry storage time (1654076) deals with synthesis and characterization of starch-silver nanoparticles and application of this coating on strawberries. Finally, strawberries characterization over time was given regarding weather coating was applied or not and weather strawberries were stored at room temperature and at cold storage.
We thank the reviewer for the clear summary of our work. We are grateful for the time and energy you expended on our behalf.
Paper’s topic is up-to-date, which is sustained with contemporary literature. Title and abstract are appropriate, so as proposed methodology. Experimental set up is clear, results well explained.
We would like to thank the reviewer for finding our manuscript carefully conducted where the results are interesting, and the discussion was well structured.
- The order of methods for starch-silver nanoparticles characterization and quality criteria of strawberry fruit should be in the same order as displayed results. Please uniform the order of methods and results to be the same.
We appreciate this suggestion; therefore, the order of methods was updated.
- Table 1 should have some kind of announcement in the text. For example: Operating conditions for… are presented in Table 1.
We appreciate the advice of the reviewer and can confirm that, it was performed.
- In part 2.2.3. (lines 139-148) provide additional info about how long immersion lasted; did you repeated the procedure or it was done once? Weather strawberry samples were kept in bulk or packed in some additional packaging material during storage period?
The missing information was mentioned in the experimental section as:
The samples were coated by the soaking method according to the method described by Ali et al.
- Change subtitle 3.2.3. to „Total soluble solid content”
We are in complete agreement with the reviewer; thus, it was changed.
Once again, I acknowledge the quality of the work and propose it for publication after minor revision.
We would like to thank the reviewer for the recommendation.
